# Transversal Kerr Effect Enhancement of Permalloy-Based Shallow Lamellar Magnetoplasmonic Crystals

Dmitry Murzin [1,*], Victor Belyaev [1], Johannes Kern [2], Corinna Kaspar [2], Wolfram H. Pernice [2,3], Rudolf Bratschitsch [2] and Valeria Rodionova [1]

[1] Research and Education Center "Smart Materials and Biomedical Applications", Immanuel Kant Baltic Federal University, 236041 Kaliningrad, Russia
[2] Institute of Physics and Center for Nanotechnology, University of Münster, 48149 Münster, Germany
[3] Kirchhoff-Institute of Physics, University of Heidelberg, 69120 Heidelberg, Germany
[*] Correspondence: dvmurzin@yandex.ru

**Abstract:** This work demonstrates spectral dependencies of reflectivity and the transversal Kerr effect of a series of permalloy magnetoplasmonic crystals based on shallow lamellar diffraction gratings with a period of 500 nm, stripe's width of 250 nm, and diffraction stripes' heights of 28, 43, 67, and 88 nm. The fabricated magnetoplasmonic crystals show a monotonic increase of the transversal Kerr effect and the diffraction figure-of-merit with higher diffraction stripes. The maximum achieved modulation value of the transversal Kerr effect was found to be 0.78%, which can be further tuned by the periodicity and stripes width of the magnetoplasmonic crystals.

**Keywords:** surface plasmon polaritons; magnetoplasmonic crystal; transversal Kerr effect; diffraction; lamellar gratings





## 1. Introduction

Research in the area of plasmonics, interaction processes between electrons in a metallic plasma, and electromagnetic radiation has led to numerous advancements in spectroscopy, nanophotonics, nonlinear optics, and sensing technologies [1–6]. The combination of classic plasmonic and ferromagnetic materials led to the emergence of magnetoplasmonics, where the plasmonic properties of a structure can be tuned via a magnetic field [7–9]. An interesting concept in this field of research is the use of magnetoplasmonic crystals (MPlCs) films made of noble and ferromagnetic metals deposited on the surface of a diffraction grating [10,11]. It is shown that MPlCs can exhibit higher sensitivity to the change of environment refraction index than usually used plasmonic crystals [12] or can be used as effective magnetic field sensing probes based, for example, on the transversal Kerr effect (TKE) detection [13]. Higher sensitivity is achieved due to the resonant enhancement of magneto-optical effects driven by the surface plasmon polaritons (SPPs) resonance interaction with the electromagnetic field inside a ferromagnetic material [8].

The excitation efficiency of SPPs on 1D corrugated surfaces strongly depends on structure periodicity, the depth of corrugation, and the shape of individual stripes [14,15]. Among frequently studied corrugated surfaces, sinusoidal and lamellar diffraction gratings are of most practical interest due to the simplicity of description and plenty of their fabrication methods. The optical and magneto-optical activity of MPlCs based on sinusoidal diffraction gratings is predominated by SPPs excitations on their surface and can be tuned by corrugation depth adjustment. The change of the stripe's height provides optimal coupling between SPPs and an incident electromagnetic wave due to the radiative and absorption losses balance [16]. On the other hand, lamellar gratings under the illumination of a *p*-polarized light support many different cavity modes and surface modes, affecting their optical properties [17–20]. It is shown that depending on stripe height, different diffraction efficiency and electromagnetic wave energy distribution can be observed due to

the complex interaction between gap plasmons, photonic modes, and SPPs between the stripes of such diffraction gratings [15]. Thus, one of the approaches to tune the enhanced TKE resonance in a lamellar MPlC is to design diffraction grating parameters like period or stripes' width and height according to the required resonance magnitude, spectral position, and spectral width. In particular, the use of shallow lamellar gratings is preferable because gap plasmons and photonic modes in them can be suppressed to avoid their overlapping with SPPs modes in a narrow spectral range [21].

This work is devoted to the experimental study of reflectivity and transversal Kerr effect spectra in the visible-NIR wavelength region of the shallow lamellar permalloy-based MPlCs depending on the diffraction stripes height. Structures period and stripe's widths were fixed to 500 nm and 250 nm, respectively.

## 2. Materials and Methods

MPlCs were fabricated in two steps by electron-beam lithography (e-beam) and magnetron sputtering methods. At first, a uniform layer of the ARP672.045 photoresist was formed by a spin coater on top of a silicon substrate at $1200 \times g$ rpm and baked at 90 °C for 2 min. The resist thickness was studied with an optical white light interferometer and was equal to 400 nm. Shallow lamellar diffraction grating patterns with a period of 500 nm and a stripe's width of 250 nm were prepared using the K-Layout software package [22]. The E-beam lithography process was done in one cycle, forming 4 identical patterns with exposure doses of 400, 600, 800, and 1000 $\mu C/cm^2$ using a Raith EBPG5150 system. After lithography, the photoresist was chemically developed in the MIBK:IPA (3:1) solution for 60 s and dried in a nitrogen gas flow. Spatial profiles of the fabricated gratings were studied with the atomic force microscope (AFM) XE-100 by Park in semi-contact mode. Figure 1a shows the AFM image of the diffraction grating with a stripe's height of 88 nm, and Figure 1b shows profiles of all the diffraction gratings fabricated on the sample.

The resulting shallow lamellar diffraction gratings with the fixed period and stripes width had the stripes' heights of $28 \pm 0.3$, $43 \pm 0.5$, $67 \pm 0.2$ and $88 \pm 0.6$ nm. Then, the DC magnetron ORION-8-UHV by AJA International was used to cover the sample surface with 50 nm of Ag, 150 nm of Py ($Ni_{80}Fe_{20}$), and 20 nm of $Si_3N_4$ at a base pressure of 6 mTorr and an argon flow of 15 sccm. The Ag layer was used to increase the adhesion of the Py layer during sputtering, and the $Si_3N_4$ layer was used to protect the sample from oxidation and scratches.

Reflectivity and TKE spectra were studied with a setup consisting of a halogen lamp, the MS3500i monochromator by Sol Instruments, a system of collimation lenses, a dichroic film polarizer, and the APD130A2 avalanche photodiode by Thorlabs as a light detector. The scheme of the setup is shown in Figure 1c. Measurements were carried out in $p$-polarized light under a light incidence angle of 45° to detect −1st diffraction order in the visible-NIR wavelength region. Since the TKE value commonly increases with the light incidence angle, the light incidence angle of 45° also allowed the measuring of moderate TKE values [23]. A Lock-In detection technique was applied for signal detection using the SR830 amplifier. For optical measurements, an optical chopper with a frequency of 244 Hz was used. For magneto-optical signal detection, a pair of Helmholtz coils generating modulation AC magnetic field along the diffraction gratings stripes with a magnitude of 250 Oe and a frequency of 68 Hz was used. The magnitude of the applied magnetic field was enough to saturate the MPlCs.

The TKE value was defined with the equation:

$$TKE(\lambda) = (R_{+H} - R_{-H})/R_0 = \Delta R/R_0, \tag{1}$$

where $R_{+H}$ and $R_{-H}$ are reflection amplitudes dependent on the modulation magnetic field direction, and $R_0$ is the reflectivity amplitude in the absence of the modulation magnetic field. The transversal Kerr effect modulation value ΔTKE was calculated as the difference between the maximum and local minimum TKE value in the wavelength region (see Figure 2d), corresponding to the SPP-assisted TKE enhancement.

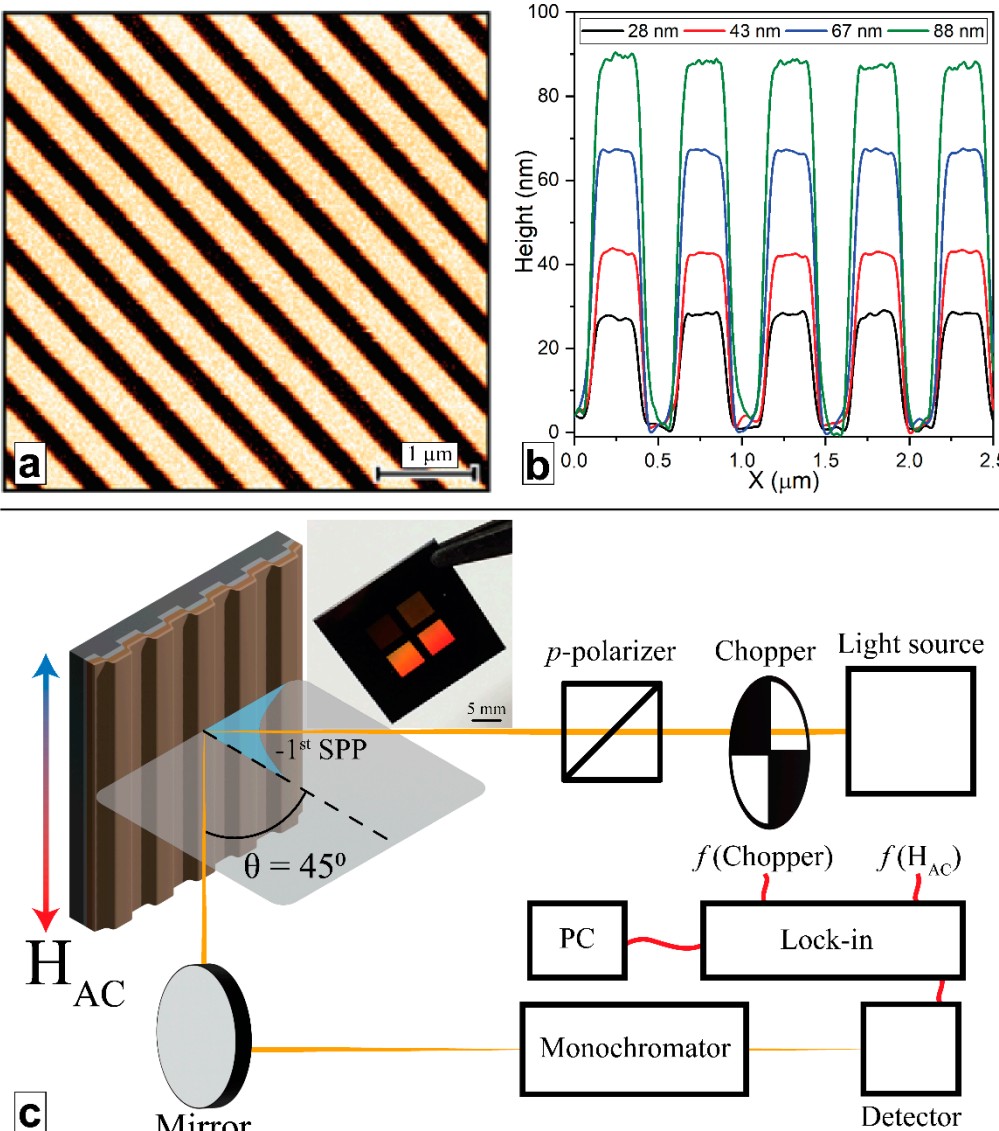

**Figure 1.** (**a**) AFM image of the diffraction grating with a stripe's height of 88 nm. (**b**) Profiles of the fabricated diffraction gratings with stripes' heights of 28 (black), 43 (red), 67 (blue), and 88 (green) nm. (**c**) Scheme of the setup used to measure reflectivity and TKE values of studied MPlCs. The angle of light incidence $\theta$ = 45°. $H_{AC}$ is the external modulation AC magnetic field applied to an MPlC along the diffraction grating stripes, and $f$ (Chopper) and $f$ ($H_{AC}$) are modulation chopper and magnetic field reference frequencies, respectively, used for the detection of optical and magneto-optical signals. The inset shows a photo of the fabricated sample containing four MPlCs.

The relative diffraction figure-of-merit (FOM) was calculated as the ratio:

$$Diffraction\ FOM = \left(I_{film} - I_{diff}/I_{film}\right)\cdot 100\%, \tag{2}$$

where $I_{diff}$ is the light intensity diffracted into the −1st diffraction order, and $I_{film}$ is the light intensity reflected from a flat unpatterned region of the sample at the same wavelength. The diffraction FOM includes not only the amount of light diffracted to the −1st diffraction order relative to the flat unpatterned surface but also optical losses due to light absorption, scattering, and coupling to SPP excitation.

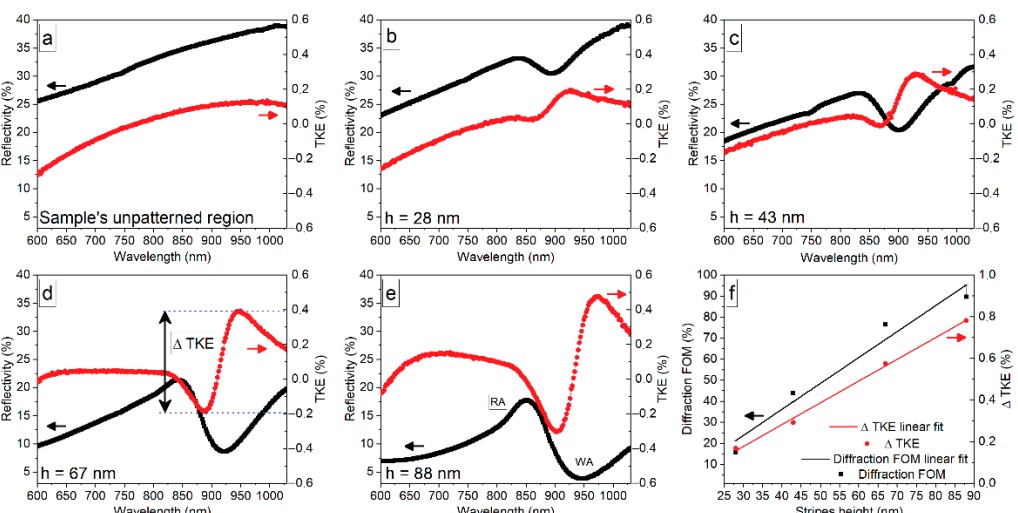

**Figure 2.** Reflectivity (black rectangles) and transversal Kerr effect (red circles) spectra for the (**a**) flat unpatterned region of the sample and the MPlCs with stripes' heights of (**b**) 28, (**c**) 43, (**d**) 67, and (**e**) 88 nm, as well as (**f**) dependencies of the MPlCs diffraction FOM (black rectangles) and transversal Kerr effect modulation value ΔTKE (red circles) on the height of stripes. Insets on panel (**e**) indicate the positions of the Rayleigh (RA) and Wood (WA) anomalies. Solid lines in panel (**f**) indicate the linear fit and corresponding residuals of diffraction FOM (black) and TKE modulation value (red), respectively. The arrows on each panel point to the *y*-axis for the nearby dependencies.

## 3. Results and Discussion

Measured visible-NIR reflectivity and TKE spectra, as well as the dependence of the TKE modulation value and diffraction FOM on the diffraction gratings stripes height, are shown in Figure 2. Reference reflectivity and TKE spectra measured from the flat, unpatterned area of the sample, having the same composition and thickness of functional layers, showed no pronounced dips or resonant features.

All reflectivity spectra of the MPlCs had a broad dip related to the evanescent light diffraction to the −1st diffraction order and subsequent coupling of the electromagnetic energy to the SPPs called Wood anomaly [24]. Such anomalies can be observed only in the case of *p*-polarized light, and their spectral position in the case of diffraction gratings without dielectric coating can be estimated according to the formula:

$$m\lambda = d\left(\sqrt{\frac{\varepsilon_1}{\varepsilon_1 + 1}} + \sin\alpha\right),\tag{3}$$

where *m* is the diffraction order, λ is the wavelength, *d* is the period, α a is the angle of light incidence, and $\varepsilon_1$ denotes the Py permittivity at the corresponding wavelength, respectively [25]. According to the Py permittivity taken from [26], the approximate spectral position of the Wood anomaly was λ = 874 nm. However, in addition to the common red shift of the Wood anomaly for dielectric overcoated gratings caused by the top dielectric $Si_3N_4$ layer [27,28], the growth of the diffraction stripe's height also led to the anomaly's slight shift to a region of higher wavelengths. More information on Wood anomalies emerging in diffraction gratings made of conducting materials can be found in Refs [24,29].

The MPlCs with the diffraction stripes' heights of 67 and 88 nm also possessed a well-pronounced Rayleigh anomaly [30]—an intensity peak next to the reflection minima related not to the excitation of surface waves but to the discontinuity in the diffracted power and following redistribution of the diffracted light energy among other propagating orders [31]. The spectral position of the Rayleigh anomaly depends only on the structure's periodicity and is defined by the diffraction grating formula:

$$\sin \alpha = \sin \beta + \frac{m\lambda}{d}, \tag{4}$$

where $m$ is the diffraction order, $\lambda$ is the wavelength, $d$ is the period, and $\alpha$ and $\beta$ are angles of reflected and diffracted light, respectively. The angle of the light incidence should be taken anticlockwise from the normal to the grating, while the angle of diffraction is taken clockwise from the normal. Coupling of the diffracted light to the propagation occurs in the wavelength region where the diffracted order appears at a grazing angle or $\beta$ equal to $90°$ [24].

Excitation of SPPs also manifested itself as the Fano-shape resonance in the TKE spectral dependencies at the wavelength region corresponding to the Wood anomalies of MPlCs. It is shown that the TKE enhancement is a consequence of the SPP nonreciprocity effect taking place in magnetic materials [32] and that in the first approximation for corrugated surfaces, it can be expressed in terms of the SPP wavevector [25]:

$$k_{spp} = k_0 \left[ \sqrt{\frac{\varepsilon_1 \varepsilon_2}{\varepsilon_1 + \varepsilon_2}} + \frac{ig\varepsilon_1^2}{\left(\varepsilon_2^2 - \varepsilon_1^2\right)\sqrt{\varepsilon_1 + \varepsilon_2}} \right], \tag{5}$$

where $\varepsilon_1$ and $\varepsilon_2$ are Py and $Si_3N_4$ permittivity, $k_0$ is a wavevector of the plane propagating wave, and $g$ is the gyrotropy. The second part of the equation represents the SPP excitation frequency shift related to the external magnetic field applied to the magnetic medium due to the gyrotropy dependence on the direction of the applied magnetic field.

The diffraction FOM and TKE values of the MPlCs had a strong dependence on the diffraction grating stripes' height. Measured values of the diffraction FOM and the TKE are shown in Figure 2f. Increase of the diffraction stripes height results in the Wood anomaly and TKE resonance broadening, as well as in the almost linear increase of TKE and diffraction FOM. The correlation coefficients were 0.980 and 0.983 for diffraction FOM and TKE modulation values, respectively. Such behavior is attributed to the SPP wavenumber's monotonic increase with the height of diffraction grating stripes observed for similar structures with small corrugation depth [21]. An increase in the MPlC corrugation depth results in the deviation of its dispersion function from the dispersion of a flat surface, resulting in the emergence of different types of surface diffuse waves, including SPPs [33]. According to the obtained results, the studied MPlCs supported only the excitation of SPPs in the visible wavelength region since no additional anomalies related to photonic or groove plasmons were seen in the experimental spectra.

The MPlC with the highest stripe's height of 88 nm shows almost 0 reflectivity at $\lambda = 948$ nm and the most profound $\Delta$TKE value. TKE values of fabricated MPlCs are about two times lower, and the spectral widths of their TKE resonant features are about two times wider than those observed for previously published results on MPlCs based on thick Py films [11]. However, fabricated lamellar MPlCs with diffraction stripes' heights of 28, 43, 67, and 88 nm showed higher enhancement of the $\Delta$TKE up to x4.9, x10.7, x28.5, and x44.5, respectively, in comparison with the flat unpatterned region having the same composition. Thus, proper adjustment of lamellar MPlC morphology parameters can be further applied to fabricate effective MPlC-based magnetometry probes, according to the protocol shown in [13].

## 4. Conclusions

In conclusion, the dependence of the diffraction figure-of-merit and the transversal Kerr effect enhancement of the shallow lamellar magnetoplasmonic crystals on the stripe's height was studied. The magnetoplasmonic crystal with a stripe's height of 88 nm showed almost zero reflectivity dip and reached the transversal Kerr effect modulation value of 0.78%. A maximal enhancement of the transversal Kerr effect modulation value with respect to the unpatterned sample region of x44.5 was shown; this is higher than our previously published results. The obtained results showed a monotonic dependence of the

diffraction FOM and the transversal Kerr effect modulation values on the stripe's height. These values can be further tuned by changing the periodicity and widths of the diffraction grating stripes to achieve the required sensitivity of the MPlCs to the changes of the external magnetic field for sensing applications.

**Author Contributions:** Conceptualization, V.B., D.M.; investigation, V.B., D.M., J.K., C.K.; resources, R.B., W.H.P., V.R.; data curation, V.B.; writing—original draft preparation, D.M., V.B.; writing—review and editing, D.M., V.B., R.B., J.K., C.K., W.H.P.; visualization, D.M.; supervision, R.B., V.R.; project administration, V.B. All authors have read and agreed to the published version of the manuscript.

**Funding:** This research was financially supported by the Ministry of Science and Higher Education of the Russian Federation, grant No 075-15-2022-272.

**Institutional Review Board Statement:** Not applicable.

**Informed Consent Statement:** Not applicable.

**Data Availability Statement:** Not applicable.

**Conflicts of Interest:** The authors declare no conflict of interest.

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
