# Peer review of "Transversal Kerr Effect Enhancement of Permalloy-Based Shallow Lamellar Magnetoplasmonic Crystals"

_photonics, doi:10.3390/photonics9120989_

Round 1

Reviewer 1 Report

The paper is devoted to study experimentally transverse magneto-optical Kerr effect (TMOKE) in 1D magnetoplasmonic crystals (MPlC). Magnetic material is permalloy. Authors take 4 samples with various height of strips in the grating. The maximal value of the TMOKE is observed for the sample with the highest strip, which is 40 times greater than that for a noncorrugated film.

The manuscript is well-written, easy to read and contains clear figures. 

However, I have some minor comments. The main one concern the definition of “diffraction efficiency” in this paper. Worldwide understanding of this term is the ratio of light power of the diffraction maxima to the incident radiation power. I don’t think it’s a good idea to name the parameter (1-I_diff/I_film) as diffraction efficiency for the following reasons. (i) Authors don’t measure the power of diffraction maximum. (ii) They measure zeroth order mode reflection. It means that the drop in the spectrum of reflection can be due to not only Rayleigh or Wood anomaly, but also by absorption (extinction) caused by resonance of surface plasmon-polariton excitation [10.1103/PhysRevB.70.125113]. (iii) It is hard to speak about the efficiency when absolute value of reflection decreasing. Although I understand what authors try to emphasize by this parameter, I recommend to use another designation for it. May be, “figure-of-merit” will be better, or they can suggest more suitable version.

The second comment concerns the Fig.2f, right axis. I did not find the definition of delta TKE. Please, introduce it in the main text of the manuscript. There is a phrase in the text (line 126): “Measured values of the diffraction efficiency and the TKE are shown in Fig 2 (f).” So, I wonder TKE and delta TKE are the same?

III.

Usually, a chopper is used after the light source to modulate incident radiation. This will increase SNR. Otherwise, it will modulate as useful signal as noises from elsewhere. Being put before the photodetector its functionality is lost.

IV.

When authors speak about enhancement in the end, it would be better to announce the reference sample. It was the film, as I understand. Otherwise, it looks like there are being compared between each other: enhancement of the 28-nm height sample with 88-nm sample, for example.

V.

Authors don’t compare their result of TMOKE with another in the literature and don’t discuss this comparison. It would be more convenient for readers to have such analytics to understand the benefits of the proposed MPlC. Please, enhance discussion section with advantages and disadvantages of the permalloy-based MPlC with other systems.

Nonetheless, I think this manuscript requires minor revisions and can be published in the Photonics after elimination of the proposed above comments. 

Reviewer 2 Report

Dear Authors,

Please, find my comments in the attached file. I recommend you to add some details and explanations to the text to make the results interpretation clearer.

Reviewer 3 Report

In this manuscript, the authors experimentally studied the spectral dependencies of reflectivity and the transversal Kerr effect of a series of permalloy magnetoplasmonic crystals with different geometrical parameters. A monotonic increase of the transversal Kerr effect and the diffraction efficiency were found in higher diffraction stripes.

Overall, this manuscript is well written and organized, and the results are of some interesting. In my opinion, this manuscript can be considered for publication after the following issues are addressed:

1, The authors wrote that the transversal Kerr effect modulation value is 0.78%. According to its definition, I suggest the authors to check if this value is 78%.

2, The quality of Fig. 2f should be improved.

3, The authors stated in the conclusion that “ ... and widths of diffraction gratings stripes to achieve required sensitivity of the MPlCs to the changes of external magnetic field for sensing applications.” However, the role of external magnetic field was not well demonstrated in this manuscript. More data or part should be provided.

4, Some references that are also concerned with stripes- and surface plasmon-based modulators maybe cite and discussed here, such as Physical Review B, 106, 075401 (2022) and "Two Switchable Plasmonically Induced Transparency Effects in a System with Distinct Graphene Resonators", Nanoscale Research Letters, 15 (1), 1-13 (2020).

Round 2

Reviewer 2 Report

Dear Authors,

Thank you for taking into consideration my remarks from the first review round. I still think that comparison with a mathematical model and some more extensive experimental study would make this paper more self-consistent and accomplished work. However, I don't see any critical flaws or lack of novelty, so the paper can be published in the current form. I also noiced that the manuscript is entitled as "Communication" now. This status also makes absense of some aspects of the research more understandable.